# *Pseudomonas aeruginosa* Clusters Toxic Nickel Nanoparticles to Enhance Survival

**DOI:** 10.3390/microorganisms10112220

**Published:** 2022-11-10

**Authors:** Ehsan Asghari, Bernhard Peter Kaltschmidt, Luis van Merwyk, Thomas Huser, Christian Kaltschmidt, Andreas Hütten, Barbara Kaltschmidt

**Affiliations:** 1Department of Cell Biology, Faculty of Biology, Bielefeld University, 33615 Bielefeld, Germany; 2Department of Thin Films and Physics of Nanostructures, Center of Spinelectronic Materials and Devices, Faculty of Physics, Bielefeld University, 33615 Bielefeld, Germany; 3Department of Biomolecular Photonics, Faculty of Physic, Bielefeld University, 33615 Bielefeld, Germany

**Keywords:** biofilm, nanoparticles, nickel, nickel oxide, *Pseudomonas aeruginosa*, *Staphylococcus aureus*, eDNA

## Abstract

Microorganisms forming a biofilm might become multidrug-resistant by information exchange. Multi-resistant, biofilm-producing microorganisms are responsible for a major portion of hospital-acquired infections. Additionally, these microorganisms cause considerable damage in the industrial sector. Here, we screened several nanoparticles of transition metals for their antibacterial properties. The nanoparticles sizes of nickel (<300 nm) and nickel oxide (<50 nm) were analyzed with transmission electron microscopy. We could show that the antibacterial efficacy of nickel and nickel oxide nanoparticles on *Pseudomonas aeruginosa* isolated from household appliances and *Staphylococcus aureus* was the highest. Interestingly, only *P. aeruginosa* was able to survive at high concentrations (up to 50 mM) due to clustering toxic nanoparticles out of the medium by biofilm formation. This clustering served to make the medium nearly free of nanoparticles, allowing the bacteria to continue living without contact to the stressor. We observed these clusters by CLSM, SEM, and light microscopy. Moreover, we calculated the volume of NiO particles in the bacterial biofilms based on an estimated thickness of 5 nm from the TEM images as an average volume of 3.5 × 10^−6^ µm^3^. These results give us a new perspective on bacterial defense mechanisms and might be useful in industries such as water purification.

## 1. Introduction

Nanoparticles have been used as antibacterial agents in medicine for thousands of years [1]. In the 20th century, organic antibiotics were discovered and replaced nanoparticles [2]. Currently, antibiotics are the standard agents used to kill bacteria. Antibiotics can damage and destroy components of prokaryotic cells that are not present in eukaryotic cells [3,4]. In general, antibiotics interfere with various mechanisms of bacteria, such as DNA replication, transcription, translation, and cell wall synthesis [5]. Nevertheless, microorganisms mutate constantly, and the mutations cause resistance to antibiotics. The basic and effective mechanism of bacteria is the production of an enzyme that can inactivate, degrade, or alter antibiotics [6]. The neurohabilitation of penicillin, the production of β-lactamase enzymes, or the cleavage of β-lactam rings are examples of bacterial adaptation to antibiotics [7]. The various modifications and mutations or transfer of a copy of DNA between microorganisms can lead to multidrug resistance (MDR) [8]. MDR is a property of bacteria that are insensitive to lethal doses of antibiotics. MDR has become a major problem for public health. Statistics show that in Europe, each year, nearly EUR 1.5 billion is spent to fight MDR microorganisms. In the USA, the expenditure on MDR amounts to USD 55 billion [9]. Moreover, more than 2 million people are infected with MRD microorganisms every year in the USA, and MDR is the cause of death for 23,000 patients [10]. Of course, the ineffectiveness of antibiotics is not the only reason why the war against microorganisms in medicine and industry is difficult. Possible reasons are the overgrowth of MDR, inadequate access to the infection area or the complex structure, and the protective mechanisms of biofilms [11]. The formation of biofilms and living in biofilms are part of the survival strategy of bacteria. The bacteria can adhere to the surface reversibly or irreversibly. If the bacteria stick to the surface, they lose pili. Then, the bacteria proliferate and secrete a self-produced matrix—extracellular polymeric substance (EPS). EPS consists of polysaccharides, glycoproteins, glycolipids, and in many cases, a large amount of extracellular DNA (eDNA) [12]. Life within the biofilm allows for a great chance of survival through various strategies of the bacteria. The bacteria in the biofilm cannot be recognized by the immune system, or the bacteria develop a strategy to survive in case of anoxia or nutrient deficiency. Under stress, the cells produce an alternative metabolism or gene expression or a protein that minimizes cell division or leads to a reduction in metabolism to consume fewer nutrients [13,14]. Another very important function of the biofilm is that the bacteria in the biofilm are insensitive to antibiotics.

In terms of antibacterial control, nanoparticles (NPs) offer an effective alternative to antibiotics to fight bacteria. NPs are not only transporters of natural antibiotics, but they actively fight against bacteria [15]. NPs can have different effects on bacteria depending on their chemical and physical properties and can damage bacteria differently [16]. Moreover, a feature that makes nanoparticles interesting as antimicrobial agents is the inability of bacteria to become resistant to them [17]. The metals that are interesting as antibacterial agents are mostly from the d-block of the periodic table, such as titanium, nickel, copper, molybdenum, silver, tantalum, gold, and a few others. Nickel (Ni) is one of the metals that has great antibacterial potential. The antibacterial effect of Ni can be influenced by the size of the NPs. NPs, which have a size of 10–100 nm, show a strong antibacterial effect [18].

In addition, researchers have been able to show, in recent years, that Ni can be used as a powerful weapon against MDR bacteria, such as *Pseudomonas aeruginosa* and *Staphylococcus aureus*.

There are few studies that have investigated the release of Ni from Ni-containing particles [19,20]. Even less research has been conducted on the micro- and nanoparticles of Ni and their toxicological effects. As an example, Pietruska et al. [21] showed the release of nickel particles from Ni-containing particles in growth medium. In addition, the toxicity of Ni nano- and microparticles and nickel oxide (NiO) nanoparticles has been studied.

In this study, we investigated the effects of transition metal nanoparticles on the growth and biofilm formation of different bacteria. Here, we cultured *P. aeruginosa* (isolated from domestic washing machines) and *S. aureus* (DSMZ 24167) in the presence of different concentrations of nickel and nickel oxide to investigate the toxicity of these metals. *P. aeruginosa* showed interesting behavior when cultivated with nanoparticles. *P. aeruginosa* was able to remove the stress factor in the medium during growth and biofilm formation and inactivate it by building a network through biofilm formation and trapping and isolating the nanoparticles from the medium. After 32 h of incubation, a floating black cluster could be observed in the medium. This cluster was the nanoparticles added to the medium at the beginning of cultivation and isolated from *P. aeruginosa* after 32 h. This unique behavior was not observed in the *S. aureus*. To better understand this process, we calculated the number of live bacteria (CFU). Furthermore, we stained the generated cluster from *P. aeruginosa* with a specific dye for eDNA and examined it with a confocal laser scanning microscope (CLSM).

Here, we detected a novel defense mechanism of *Pseudomonas aeruginosa* against toxic NPs by clustering the NPs in biofilm.

## 2. Materials and Methods

### 2.1. Bacterial Species 

For this study, we used two bacterial species: a Gram-negative strong biofilm producer *P. aeruginosa*, which was isolated from household washing machines in the previous work and identified by MALDI-TOF MS [12], and *S. aureus*, which we ordered from the German Collection of Microorganisms and Cell Cultures (DSMZ 24167). Bacteria were stored at −80 °C and grown overnight in liquid LB (Luria broth) medium at the temperature of 37 °C for all experiments. Overnight culture was also cultivated on LB agar plates and incubated by 37 °C.

### 2.2. Nickel and Nickel Oxide Particles

Several nanoparticles were tested as part of this study (see Table 1). All nanoparticles in different concentrations were tested on *P. aeruginosa* and *S. aureus* on LB-Agar plates. Then, we decided to focus on two nanoparticles (nickel and nickel oxide) because their bactericidal effects were strongest. We ordered all nanoparticles from Sigma Aldrich (Sigma-Aldrich Chemie GmbH, Taufkirchen, Germany) as nanopowders except for the micro-silver, which we ordered from Biogate (Bremen, Germany). In this work, since our focus was only on nickel and nickel oxides, we only present the manufacturer information (size of the nanoparticles) in comparison with our measurements with TEM in Table 2.

### 2.3. Particle Morphology and Size

To analyze the particle size and morphology of the nickel and nickel oxide nanoparticles, a transmission electron microscope (Jeol 2200 FS, Akishima, Tokyo, Japan) was utilized. To prepare the samples for microscopy, aliquots were first dissolved in toluene, and then, 1 µL of the solution was deployed on a copper TEM grid (Plano GmbH, Wetzlar, Germany) with a pipette. The excess liquid was then drawn off with filter paper. After 15 min of drying, the samples were analyzed in the microscope at an acceleration voltage of 200 kV.

### 2.4. Determination of Minimum Inhibitory Concentration

To determine the minimum inhibition concentration of the NPs, agar plates were prepared with different concentrations of nanoparticles (5, 10, 25, 50, 100, 250, and 500 mM). Afterwards, the bacteria were spread from the overnight culture with an inoculating loop on an agar plate and incubated at 37 °C for 24 h. The next day, the plates were visually examined and determined to what concentration the bacteria could grow in the presence of nanoparticles.

### 2.5. Colony-Forming Unit Assay (CFU) to Analyse Metal Toxicity

The colony-forming unit assay (CFU) was performed as a quantitative method for the determination of live bacteria. For this assay, an overnight culture was prepared in 10 mL LB medium at 37 °C. Different concentrations of Ni and NiO (5 mM, 10 mM, 25 mM, and 50 mM) were prepared in 20 mL of fresh LB medium. Then, the media with different concentrations of nanoparticles were inoculated to OD_600_: 0.05 and incubated at 37 °C and 120 rpm. The colony-forming units were determined immediately after inoculation and after 1, 2, 4, 8, 16, and 32 h. Serial dilutions were performed in 5 mL of physiological saline (0.9% NaCl), and 100 µL was added to the LB agar plates. The plates were incubated at 37 °C for 24 h, and the colonies were counted the next day. In this study, all tests were performed in triplicate.

### 2.6. Scanning Electron Microscopy

To better understand the interaction of nanoparticles and bacterial surface or biofilm, both bacteria were observed using a scanning electron microscope. As described in the CFU method, bacteria were cultured with different concentrations of nanoparticles in LB medium. For SEM, a 50 µL sample of 5 and 50 mM medium containing the aforementioned nanoparticles was taken immediately after the addition of bacteria and after 32 h of cultivation and pipetted onto a coverslip. Due to the high salt concentration in the medium, the salt was highly crystallized after drying. The crystallization limited the observation of the surface of the bacterial biofilm. To address the issue, the samples were placed in a Petri dish, ddH_2_O was gently added, and they were incubated at room temperature for 30 min. The water was then removed and dried at room temperature. To ensure a conducting surface and thus, the minimization of charging effects, all samples were sputter-coated with a 4 nm layer of Ruthenium prior to microscopic analysis. The ruthenium-coated surface was also conductively connected to the sample holder with the use of carbon tape. For analysis, an acceleration voltage of 5 kV and a beam current of 0.17 nA were chosen.

### 2.7. Confocal Laser Scanning Microscopy

The samples were examined with a confocal scanning microscope to determine and localize the biofilm. The dye TOTO-1 (Life Technologies Corporation, Eugene, OR, USA) was used to stain the eDNA and biofilm. TOTO-1 is a non-permeable dye that stains only eDNA and is therefore optimized for biofilm staining. In combination with TOTO-1, we dyed all of the bacteria with Syto™60 (Molecular Probes, Inc, Eugene, OR, USA). Syto™60 is a membrane-permeable, nucleic acid-staining, red fluorescent dye, which is used as a counterstain. The excitation/emission maxima reported by the kits are 514/533 nm for TOTO-1 and 652/678 nm for Syto™60. The samples for CLSM were also cultured as described for CFU. For the CLSM, we collected and stained the clusters of nanoparticles from mediums containing 5 mM and nickel oxides, which were formed by *P. aeruginosa.*

### 2.8. Light Microscopy

For light microscopy, we used an Axiophot (Zeiss company, Oberkochen, Germany). The formed cluster of *P. aeruginosa* was taken out of the medium containing 5 mM nickel oxide and washed 2 times with PBS. The cluster was then placed on a microscope slide. The cluster was viewed with a 10× objective. The size of the cluster was measured by Image J and analyzed statistically. This statistical analysis was performed using 3 images, and 12 clusters were measured from each image.

### 2.9. Light Sheet Microscopy

For light sheet microscopy, custom-made sample chambers were 3D-printed (Prusza) using PLA. A glass plate was bonded to the bottom of each chamber. Then, 10 mL of LB medium with 5 mM nickel oxide was prepared to OD600 = 0.05 and injected into the chamber. The chamber was sealed with the help of a plastic cover so that the medium would not spill during the shaking in the incubator. The samples were incubated for 8, 16, and 32 h at 37 °C and 120 rpm. After incubation, the chambers were washed 2× with 1% PBS and fixed with 4% PFA for 30 min to inactivate the bacteria. Afterwards, the chamber was washed 2 more times with 1% PBS and air-dried. Imaging was done using a custom-made inverted light sheet microscope. To create the light sheet, a Bessel beam was scanned and a special excitation objective (Thorlabs TL20X-MPL) was used to project the sheet into the sample. The collection of the fluorescence light was done with a 25× 1.05 NA objective (Olympus-XLPLN25XWMP2), a 250 mm tube lens to roughly achieve a 34× magnification, and a CMOS camera (IDS-µeye 3270). Sample movement was achieved with a closed-loop linear piezo stage (Piezosystem Jena, nanoSX 800). The image stacks were fed into a simple but custom-made algorithm to create a single image.

### 2.10. Data Analysis

To perform all statistical analyses, a one-way ANOVA rank test with a *p* value in GP style was performed with: 0.1234 (ns), 0.0332 (*), 0.0021 (**), 0.0002 (***), and <0.0001 (****). The error bars indicate the standard deviation (SD) for all experiments.

## 3. Results

We cultured *P. aeruginosa* (isolated from domestic washing machines) and *S. aureus* (DSMZ 24167) in the presence of different concentrations of several NPs to investigate the toxicity of these metals (Table 1).

We observed that nickel was the most efficient for killing bacteria on agar plates. Therefore, we decided to compare the toxicity of nickel and its derivatives. Thus, we characterized the nickel-containing NPs in further detail. 

First, we determined the morphology, size, and oxygen distribution on the surface of the used nickel nanoparticles by transmission electron microscopy (TEM). Figure 1 shows the morphology of the Ni NPs. The tested Ni nanoparticles had a round shape of different sizes. The size of the Ni NPs reported by the manufacturer was <100 nm (Table 2). However, according to our observations and evaluations, we found a size of <300 nm for the Ni NPs. In addition, the surfaces of the Ni nanoparticles were found to contain oxygen, as analyzed by energy dispersive X-ray spectroscopy (EDX), indicating that the surfaces of the nanoparticles were oxidized (Figure 1E). In contrast to the Ni, the NiO had triangular shapes (Figure 2). This shape of nickel oxide nanoparticles has also been observed in other studies [22]. In the nickel oxide NPs, the oxygen was equally distributed as expected by its chemical nature. The size of the NiO NPs was around 50 nm. In this case, the size of the nanoparticles is consistent with the size reported by the manufacturer (Table 2).

The number of live cells was determined during the cultivation of the bacteria with different concentrations of nickel and nickel oxide by measuring the colony formation (CFU). CFU assays were performed at the time points of 0, 1, 2, 4, 8, 16, and 32 h. *S. aureus* showed no sensitivity to a low nickel oxide concentration, and after 32 h, there was no significant cell count reduction compared to the control. However, at higher concentrations (25 and 50 mM), there was a significant reduction in the cell number after 32 h. Here, no colonies grew on the plate (Figure 3A). On the other hand, nickel showed high toxicity for *S. aureus* even at lower concentrations. At a concentration of 5 mM, the number of colonies after 32 h was almost the same as at the beginning but despite this, there was no significant reduction in the number of cells as compared to the reference. This indicates that the number of divided bacteria and dead bacteria remained almost the same. When the Ni concentration was doubled, the bacteria were completely eliminated after 16 h. At higher nickel doses, such as 25 and 50 mM, no growing bacteria were detected after 8 and 4 h, respectively (Figure 3C). In contrast to *S. aureus*, *P. aeruginosa* showed low sensitivity in the presence of NiO. Even the high concentrations of NiO could not terminate *P. aeruginosa*. After 8 as well as after 16 and 32 hours, a significant reduction of cells was observed in the medium with 50 mM NiO NPs compared to the control. However, the bacteria were not completely dead. Even in the presence of 50 mM NiO, a slight reduction in the bacterial count was observed, but the difference compared to the control was very small (Figure 3B). *P. aeruginosa* showed a provocative behavior in the presence of Ni in small amounts. At 5 mM Ni, after 16 h, the division of bacteria still took place and remained in the logarithmic phase. At 10 mM Ni, the *P. aeruginosa* had a slight increase, reaching the end point of the control after 32 h. At 25 mM, a moderately significant decrease in bacterial counts was observed after 8 h. After 32 h, the number of living cells was reduced by a factor of 10. The highest nickel concentration used in this study (50 mM) showed a toxic effect, so that after 4 h, a significant decrease in the cell number was observed and after 16 h, no growing cells were detected (Figure 3D).

In parallel to the CFU assays, the cultures were observed, and for photographic documentation, we transferred them to Petri dishes. Figure 4 shows the cultured *Pseudomonas aeruginosa* in LB medium with different concentrations of nickel and nickel oxide after 32 h of incubation at 37 °C and 120 rpm in the bacterial shaker. When *P. aeruginosa* was cultured in LB medium without nanoparticles (Figure 4A), a color change from yellow to dark green was observed. This color change was different for 5 mM and 10 mM nickel oxide. At these concentrations of nanoparticles, the medium had a neon yellow color. In addition to the color change, another remarkable property was observed in the *P. aeruginosa* cultures. The *P. aeruginosa* was able to filtrate the nickel oxide from the medium even at higher concentrations. At 5 mM, the nickel oxide was visible as a floating black cluster, while the rest of the medium was nearly free of nanoparticles (Figure 4B). Even at 10 mM, most of the nickel oxide was isolated, and only a few small clusters were additionally visible (Figure 4C). In the Erlenmeyer flasks containing 25 mM nickel oxide, a large cluster with several small clusters was identified, indicating that even at this concentration, a process of biofilm formation and isolation had occurred (Figure 4D). In the case of the highest nickel oxide concentration used in this study, no cluster was observed (Figure 4E), but the CFU results clearly show that the bacteria were not completely eliminated even at this concentration (Figure 3B). In contrast to the nickel oxide, the cluster formation was much lower with the nickel. Even at lower concentrations of nickel, very small clusters were seen, which suggests that complete filtration of the Ni NPs did not occur (Figure 4F). The same was true for 10 mM (Figure 4G). The higher nickel concentration (25 mM) had high toxicity, and the cell survival and biofilm formation were completely excluded (Figure 4G).

Seeing that the *P. aeruginosa* could form clusters in the presence of nickel nanoparticles led us to study this cluster more in detail. For this purpose, we collected the cluster from the medium of *P. aeruginosa* with 5 mM nickel oxide, washed and fixed it, and observed it with a scanning electron microscope. In addition, samples were taken from the media of *P. aeruginosa* and *S. aureus* and observed with the microscope. The observations of the cluster clearly demonstrate the interspersed structure with a considerable number of holes and channels. This form of the biofilm is quite typical for *P. aeruginosa* (Figure 5A–D). Next, we analyzed a sample taken from the medium of *P. aeruginosa* (Figure 5E). Here, we observed that a very small amount of particle residue of NiO remained in the medium. This small particle residue demonstrated that most of the nanoparticles were filtered out of the medium (Figure 5F–H). Furthermore, we examined the medium of *S. aureus* (Figure 5I). In SEM, the agglomerations of nanoparticles in the medium of *S. aureus* were clearly visible (Figure 5K) and were larger in comparison to the nanoparticle agglomerations in the medium of *P. aeruginosa* (Figure 5G). The nanoparticle agglomeration size was much larger in the medium of *S. aureus* (Figure 5K) compared to that of *P. aeruginosa* (Figure 5G,K), but no large clusters (Figure 5B) were visible in the culture of *S. aureus*. This means that the *P. aeruginosa* was more efficient in the clustering of nanoparticles and filtering the medium.

To investigate whether the clusters formed by *P. aeruginosa* contained NiO NPs, we performed an EDX analysis. We analyzed the region of Figure 5D by EDX in detail. The results of the EDX analysis show that the cluster was clearly composed of nickel and oxygen (Figure 6A). The small amount of ruthenium was due to the sputter coating to increase conductivity. Since the sample was prepared on a cover glass, a small amount of silicon was also found in the EDX analysis (Figure 6A).

For fast analysis in industrial applications, the preparation of SEM and CLSM samples, including fixing, staining, and in the case of SEM, conductive coating, are too cumbersome.

Therefore, we tested whether the unstained biofilm could be observed by a novel fast and easy-to-use light sheet microscope. To observe the cluster formation by *P. aeruginosa*, we also tested the biofilm formation on the glass surfaces, exploiting the autofluorescence. Thus, we cultured *P. aeruginosa* once without NiO and once with 5 mM NiO. The coverslips were observed with light sheet microscopy after 8, 16, and 32 h. The biofilm on the coverslips that were in the medium without NiO could be observed at all three time points (Figure 7A–C). However, compared to the coverslips cultured in the medium containing 5 mM NiO, the biofilm was much more widespread. At all three time points, it can be seen that the biofilm production became much more aggressive when the *P. aeruginosa* was cultured with NiO. In addition, after 32 h, it can be seen that the entire surface was covered with biofilm. (Figure 7D–F). This observation again confirms the hypothesis that *P. aeruginosa* showed an enormous provocation behavior when this bacterium came into contact with a stress factor such as NiO in a non-lethal concentration (as can be seen in Figure 3, showing the CFU assay).

The light microscope images of the clusters generated by *P. aeruginosa* revealed that the large clusters were composed of many small clusters. These small clusters had different sizes. After analyzing the cluster size, it was determined that most of the clusters were between 210 and 300 µm in size (Figure 8). This size range contained about 55 percent of all clusters.

Next, we used confocal laser scanning microscopy (CLSM). For the CLSM, the samples were stained with two dyes. TOTO-1 is an excellent dye for staining extracellular DNA (eDNA) (green). eDNA is one of the most important components in biofilm and serves as an adhesive in the biofilm structure. We used Syto™60 as a counterstain for staining whole bacteria, both live and dead (red). Figure 9A shows the structure of the clusters formed by *P. aeruginosa* when cultured with NiO. The different sizes of the clusters can also be seen in this panel. The black core in this image depicts the collected nanoparticles. The rings of the living and dead bacteria around the nanoparticle cores can be seen in red. These bacteria filtered the NPs from the media by biofilm production. The biofilm layer completely protected the other bacteria so that the bacteria no longer came into contact with the nanoparticles (Figure 9A). Panel B shows the 3D structure of the biofilm in Figure 9A. In this image, it can be clearly seen how the biofilm was formed from the outside around the enclosed nanoparticles by the bacteria, isolating them from the outside world. It is notable that the Ni clusters were much smaller in size than the NiO clusters. The size of the formed Ni clusters was about 50 to 60 µm. The structure of the clusters was basically the same as in the NiO. Since the clusters formed were much smaller than those of NiO, the detailed structure could not be determined by CLSM (Figure 9C). Furthermore, we statistically analyzed the clusters formed by *P. aeruginosa* in the medium containing 5 mM nickel oxide NP. The results show that over 40% of the clusters have a biofilm layer thickness of 20–30 µm. Overall, the biofilm layer thickness does not show a strong significance, only the layer thickness of 20 to 30 µm has a minor significance compared to very thin (10 to 20) and very thick layers (60 to 70 and 70 to 80 µm). In contrast to biofilm, the layer thickness of live and dead bacteria is in the range of 30 to 40 µm. This also shows a clear difference to the thinner and very thick layers. 

To estimate the number of nanoparticles in the clusters at an NP concentration of 5 mM we made the following assumptions: the NiO particles were approximately triangular prisms; therefore, we calculated an average base area of 707 nm^2^ and estimated a thickness of 5 nm from the TEM images to calculate an average volume of V_NiOP_ = 3.5363 × 10^−6^ µm^3^ for a NiO particle. The clusters of the NiO particles had a spheroid form, and we calculated an average spheroid volume of V_SNiO_ = 4.1157 × 10^6^ µm^3^. In this calculation, the thickness of the bacterial edge layer was considered. According to Kim et al., triangular prisms can form hexagonal superstructures with a packing density between 81% and 100%. In our case, the nanoprisms varied greatly in size, so we assumed a lower factor of around 80%. This means that 80% of the spheroid volume was filled with nanoparticles. Thus, we obtained an average number of 9.31 × 10^11^ nanoparticles per spheroid.

At a concentration of 5 mM, there were 0.37345 mg/mL NiO nanoparticles in the medium when assuming a homogeneous distribution. Next, we calculated the average mass of a spheroid by using a density of ρ_NiO_ = 6.72 g/cm^3^ for NiO to obtain the volume of medium in which these NPs were distributed at the start of cultivation. After the calculation, we could assume an average volume of 0.06 mL. Then, we computed the radius of a sphere with the same volume to visualize in which radius the bacteria compressed the NiO NPs. The average radius of the sphere was 2.418 mm.

Thus, we can conclude that the *P. aeruginosa* compressed the NiO NPs to an average volume of 0.004 µL, which were previously dispersed in an average volume of 0.06 mL. This is a reduction in volume by a factor of 15,000.

The Ni NP clusters resembled spheres, but because they were much smaller, we could not visualize the inner volume of the spheres by CLSM and, thus, could not measure the thickness of the outer bacterial layer surrounding the NPs. Thus, we estimated it to be around 5% of the average diameter after comparing the thickness of the bacterial layer of the NiO NPs to their spheroid volume. Therefore, we obtained an average cluster volume of V_NiS_ = 33,510 µm^3^ for the Ni clusters. The volume of the spherical Ni NPs was V_NiP_ = 0.0175 µm^3^. The highest packing density for spheres is the closest packing of spheres that amounts to 74%. Thus, we computed the average amount of Ni NPs per cluster to be 1.42 × 10^6^ particles. We then calculated the average mass of a cluster using a density of ρ_Ni_ = 8.902 g/cm^3^ for Ni and calculated the volume in which this mass of NPs was distributed (5 mM = 0.29347 mg/mL for Ni NPs) and we obtained an average volume of 0.752 µL. This is the volume of a sphere with a radius of 0.564 mm. 

Therefore, we can say that the bacteria compressed the Ni NPs, which were previously dispersed in an average volume of 0.752 µL, to an average volume of 3 × 10^−5^ µL. This is a reduction in volume by more than a factor of 25,000.

## 4. Discussion

Numerous metal nanoparticles, such as silver, titanium, copper, zinc and nickel, have been shown to be effective elements against bacteria, viruses, fungi and eukaryotic cells [22,23]. These elements have been used by humans to fight infections for thousands of years [1]. Some metals have also been demonstrated to be effective in destroying bacterial biofilms [24,25,26]. Nickel is not known for its pure form. Nickel is commonly used in the fabrication of stainless steel and other nickel-containing alloys. The use of nickel makes compounds more heat-stable and corrosion-resistant. These properties make nickel-based alloys attractive for food processing, medical devices and implants (hypodermic needles, gynecological intrauterine devices, nut bar implants, and implantable cardioverter defibrillators), and, certainly, for the chemical industry [27,28]. The antibacterial effect of nickel and nickel oxides against *Pseudomonas aeruginosa* and *Staphylococcus aureus* and many other bacteria has already been tested and determined to be highly toxic [29,30,31] but in this study we could show a new rescue mechanism of *P. aeruginosa* against toxic nanoparticles. Interestingly, a study by Perrin et al. showed that the application of Ni nanoparticles altered the approach of E. coli towards biofilm formation. The nanoparticles induced E. coli to form a biofilm instead of growing in planktonic form on the surface [32].

Gram-negative bacteria are among the most important pathogens of nosocomial infections in the medical field. Such infections can pose a risk of prolonged hospitalization and even patient fatalities. *P. aeruginosa* is a Gram-negative bacterium that is very susceptible to genetic changes. The constant genetic changes result in the bacterium being resistant to several antibiotics. This ability aids this bacterium to survive in harsh environments. Therefore, *P. aeruginosa* is one of the most significant hospital infections and also a suitable model for control strategies against multidrug-resistant bacteria [33]. In addition, *P. aeruginosa* is one of the most common biofilm-forming bacteria. In our previous study, we showed that the wild-caught *P. aeruginosa* from household appliances can form the most biofilm by far [12]. The cell wall of Gram-negative bacteria mainly consists of an outer lipopolysaccharide layer and a thinner peptidoglycan layer. This structure generates a strong negative charge on the surface of Gram-negative bacteria [34]. The surface charge may explain the antibacterial effect of nickel on *P. aeruginosa* at higher concentrations. The negatively charged surface of bacteria and positively charged nanoparticles lead to electrostatic attraction. Nickel oxides have a lower to neutral charge on the surface compared to nickel, which explains the low antibacterial effect. Furthermore, *P. aeruginosa* filters the NiO nanoparticles by biofilm formation. These phenomena lead to the fact that NiO displays the absence of any particular antibacterial effect on *P. aeruginosa*. Interestingly, NiO at concentrations of 25 and 50 mM showed high toxicity against *S. aureus* despite a neutral charge, whereas the cell wall charge is lower in Gram-positive bacteria. The size of the nanoparticles is also a parameter to be considered. The Ni NPs used in this study were >300 nm in size. Compared to the NiO NPs, which were much smaller than the Ni NPs (>50 nm), it can be concluded that the Ni NPs were much heavier and settled faster during cultivation. On the other hand, the nickel oxide nanoparticles were much smaller and also lighter, which is why they remained in the medium and did not settle. This allowed the nickel oxide NPs to be trapped by the bacteria. The size of the nanoparticles is inversely correlated to the clusters formed. NiO NPs are six times smaller than Ni NPs, but the formed clusters of NiO NPs are about five to six times larger than the clusters of Ni NPs. Metal nanoparticles are very often associated with reactive oxygen species (ROS). The ROS generated on the bacterial membrane cause membrane damage and lead to the death of bacteria [35,36]. In this study, we demonstrated that nickel at low concentrations and nickel oxide at relatively high concentrations (25 and 50 mM) can lead to the killing of *Staphylococcus aureus*. In contrast to *S. aureus*, *Pseudomonas aeruginosa* can develop a high tolerance to nanoparticles through biofilm formation. *Pseudomonas aeruginosa* can not only survive in very high concentrations of nickel oxide, but through an unknown mechanism, they can filter the stress factor—in this case, nickel oxide nanoparticles—almost completely from the medium. Therefore, *Pseudomonas aeruginosa* shows a high survival rate against nickel nanoparticles. In addition, *P. aeruginosa* can also survive at quite high concentrations of Ni again by filtering the Ni nanoparticles. Only at the highest Ni concentration (50 mM) tested in this study were we able to eliminate *P. aeruginosa* after 32 h of cultivation.

## 5. Conclusions

In this study, we showed that *P. aeruginosa* has developed an as-yet undescribed defense mechanism against toxic nanoparticles. This is possibly due to filtering the toxic NPs from the medium by forming these into dense clusters surrounded by biofilms with a compression factor of 25,000. Thus, this mechanism might be employed for the filtering of other NPs with different sizes and shapes and, therefore, could be used in the future for water decontamination or water purification. *P. aeruginosa* is one of the MDR bacteria that cause clinical infections. The analysis of this behavior may lead us to reconsider the combat strategies against this bacterium in hospital infections.

## Figures and Tables

**Figure 1 microorganisms-10-02220-f001:**
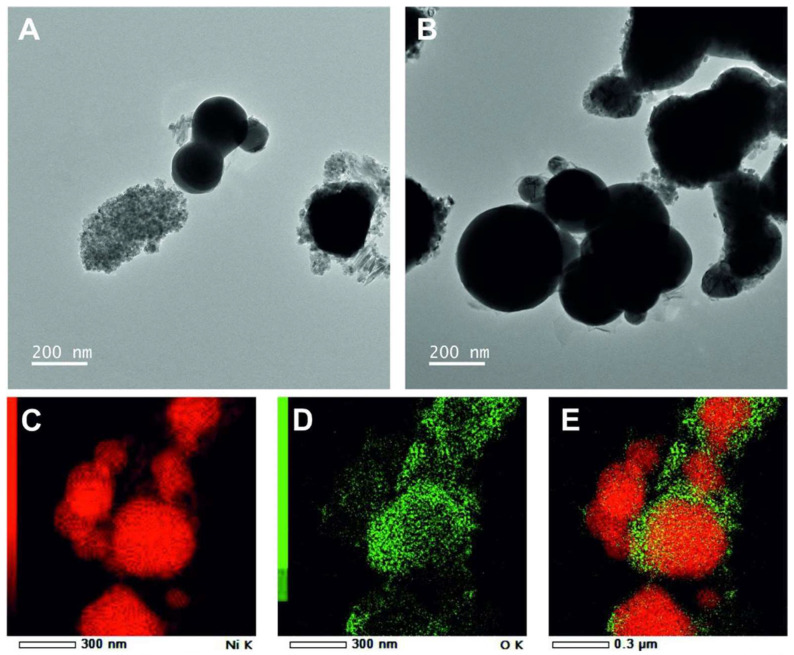
Analysis of nanoparticles and dot-mapping of nickel by transmission electron microscopy (TEM). The upper panel presents the morphology of nickel (**A**,**B**). The lower panel shows the dot-mapping of nickel and oxygen (**C**–**E**). Dot-mapping of nickel (**C**). Dot-mapping of nickel oxide (**D**). The merged representation of nickel and oxygen (**E**). This panel shows the oxygen distribution on the surface of nickel nanoparticles.

**Figure 2 microorganisms-10-02220-f002:**
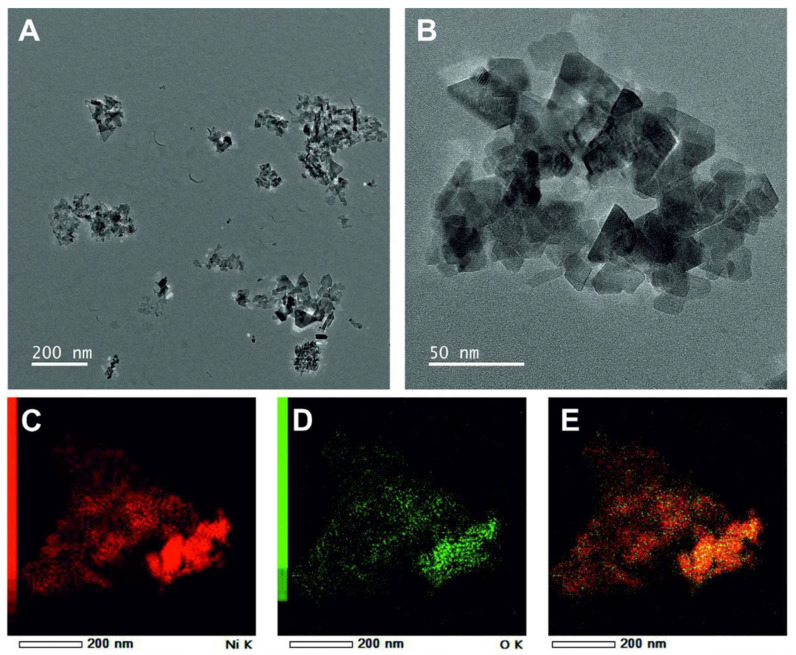
Analysis of nanoparticles and dot-mapping of nickel oxide by transmission electron microscopy (TEM). The upper panel presents the morphology of nickel (**A**,**B**). The lower panel shows the dot mapping of nickel and oxygen (**C**–**E**). Dot-mapping of nickel (**C**). Dot-mapping of nickel oxide (**D**). The merged representation of nickel and oxygen (**E**). This panel shows the oxygen distribution on the surface of nickel oxide nanoparticles.

**Figure 3 microorganisms-10-02220-f003:**
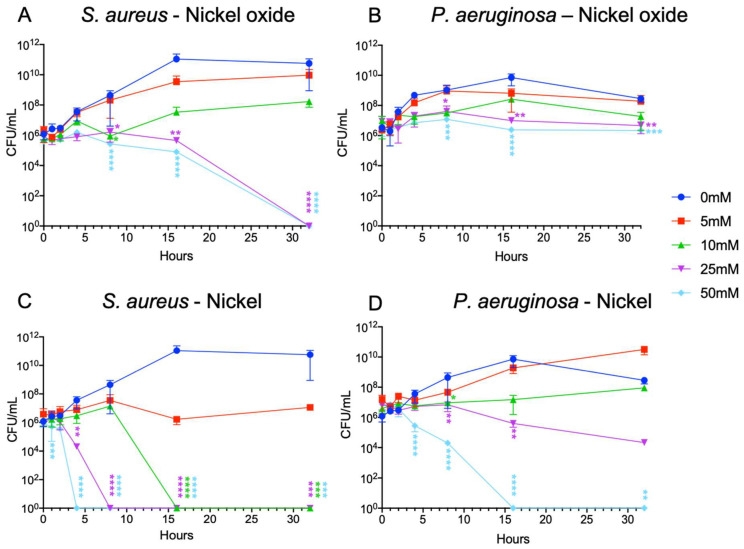
Time kill assay of *Staphylococcus aureus* (**A**,**C**) and *Pseudomonas aeruginosa* (**B**,**D**) cultured with different concentrations of nanoparticles. *S. aureus* could survive in low concentrations (5 and 10 mM) of nickel oxide without a significant effect, but at a nickel oxide concentration of 25 mM or more, growth inhibition was observed, and after 32 h, all bacteria were dead (**A**). In contrast to *S. aureus*, *P. aeruginosa* could survive at higher concentrations of nickel oxide. The bacteria entered the exponential growth phase for the first 8 h, but then cell division was reduced and the bacteria still survived (**B**). Nickel showed high toxicity to *S. aureus* at 10 mM, and the bacteria were eliminated after 16 h. At higher concentrations, e.g., 25 mM, the cells were dead after 8 h, and at 50 mM, after 4 h (**C**). *P. aeruginosa* showed provocative behavior in the presence of nickel at low levels. At 5mM nickel, after 16 h, the division of the bacteria still took place and remained in the expansion phase. At 10 mM, *P. aeruginosa* had a slight increase and reached the end point of the control after 32 h. At 25 mM, a decrease in the number of bacteria was observed after 8 h. After 32 h, the number of live cells was reduced by a factor of 10. The highest nickel concentration used in this study (50 mM) showed a toxic effect, such that a significant decrease in the cell number was observed after 2 h, and all cells died after 16 h (**D**). Definition of statistical significance: GP: 0.1234 (ns), 0.0332 (*), 0.0021 (**), 0.0002 (***), 0.0001 (****).

**Figure 4 microorganisms-10-02220-f004:**
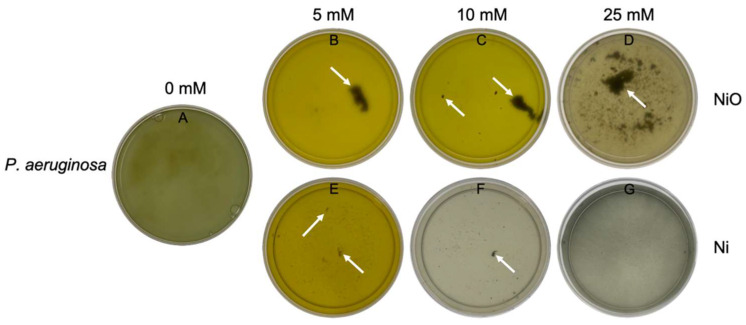
Morphology of the growth medium and formation of clusters. *P. aeruginosa* in LB medium without nanoparticles as control (**A**). *P. aeruginosa* was capable of filtering almost all of the NPs within a cluster at the low nickel oxide concentration (**B**,**C**). At the higher concentration (**D**), the accumulation was divided into several clusters. Nevertheless, most of the NPs were located within the clusters. Panels (**E**,**F**) also show the clustering of *P. aeruginosa* in Ni, but at a much smaller scale. No cluster was visible in 25 mM Ni (**G**). The images with the 50 mM concentration are not shown here because the medium was very dark due to the high concentration of NPs, making it impossible to recognize any visible features.

**Figure 5 microorganisms-10-02220-f005:**
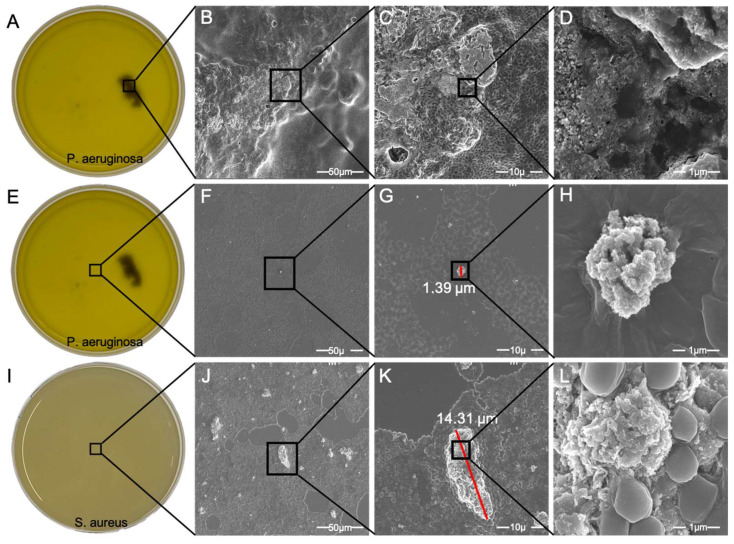
Scanning electron microscope (SEM) observation of the cluster (**A**–**D**) and medium of *P. aeruginosa* (**E**–**H**) and medium of *S. aureus* (**I**–**L**). The images show the structure of the biofilm with slime and channels (**B**–**D**). An EDX measurement was performed using photo D to analyze the content of the cluster (Figure 6). Comparing the images of the middle (**F**–**H**) and bottom (**J**–**L**) rows, it can be seen that the nanoparticles in the medium containing *P. aeruginosa* were almost completely isolated from the medium, and only very small particles remained. In contrast, in the medium with *S. aureus*, the nanoparticle dispersal was much higher and more widespread.

**Figure 6 microorganisms-10-02220-f006:**
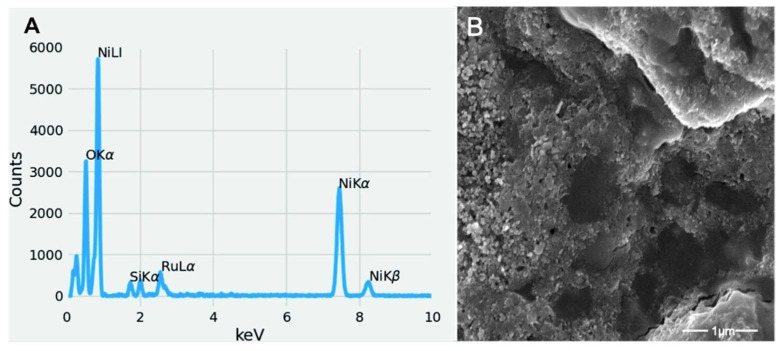
EDX analysis of the cluster formed in a medium containing 5 mM nickel oxide NPs and *P. aeruginosa* after 32 h of cultivation. The results of EDX analysis show that the cluster formed consisted mainly of nickel and oxygen (**A**). SEM enlargement of cluster (**B**).

**Figure 7 microorganisms-10-02220-f007:**
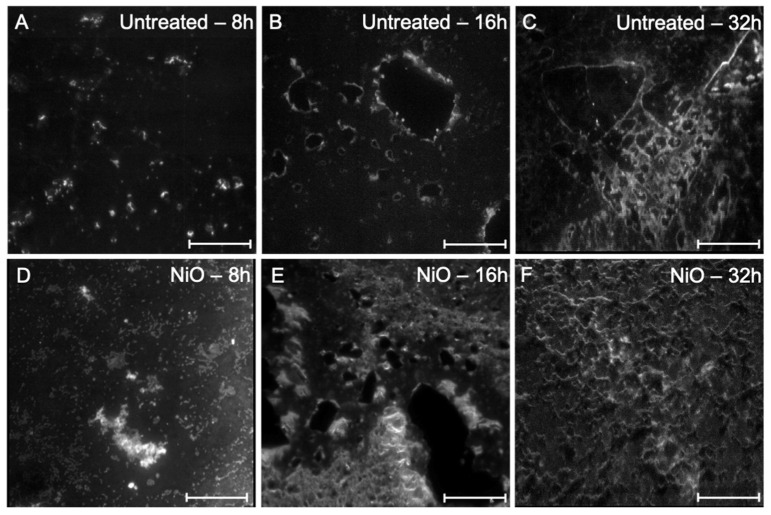
Light sheet microscopy images of *P. aeruginosa* biofilm formation on coverslips. Panels (**A**–**C**) show the biofilm formation at different time points in the LB medium alone. Panels (**D**–**F**) show the biofilm formation in the presence of NiO NPs on the coverslip in LB medium. The direct comparison at different stages clearly demonstrates that the *P. aeruginosa* formed a more drastic biofilm in the presence of NPs. Scalebar = 500 µm.

**Figure 8 microorganisms-10-02220-f008:**
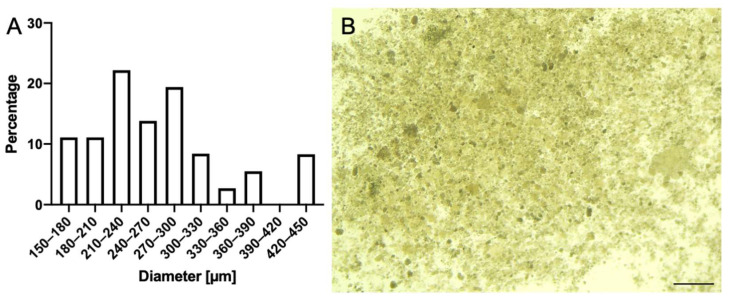
Observation of clusters from the medium with *Pseudomonas aeruginosa* containing 5 mM nickel oxide NPs using light microscopy and statistical analysis of the cluster size. The *X*-axis depicts the size of the cluster and the *Y*-axis shows the relative percentage (**A**). Cluster formation of nickel oxide (**B**). Note that the majority of the clusters were between 210 and 300 µm in size. Scalebar = 1 mm.

**Figure 9 microorganisms-10-02220-f009:**
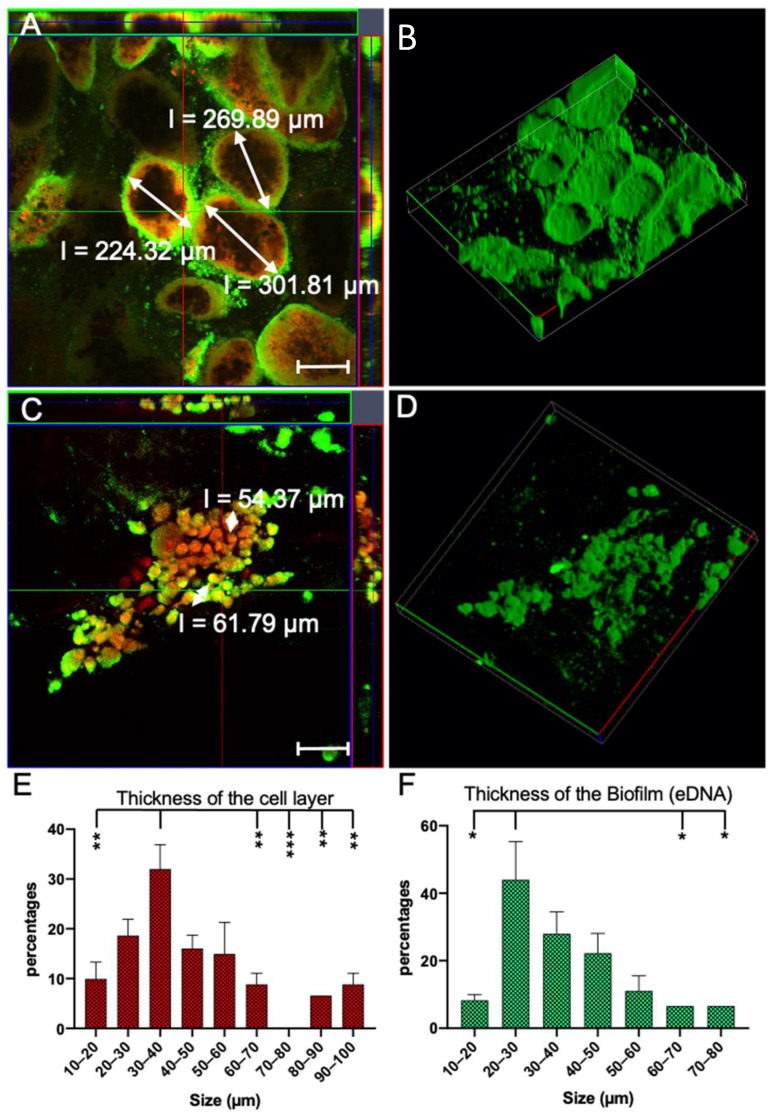
Confocal laser scanning microscopy of clusters formed by *Pseudomonas aeruginosa* in the presence of nickel and nickel oxide nanoparticles after 32 h. For this assay, the dye TOTO-1 was used for staining the eDNA biofilm (green) and Syto™60 as a counterstain (red). The cluster formed in the medium with nickel oxide had an average diameter of 200 to 300 μm. Furthermore, the bacteria clustered around the nanoparticles and formed a biofilm towards the outside. In this mechanism, some bacteria sacrificed themselves and formed a biofilm so that other bacteria survived without contact with the nanoparticles in the medium (**A**). The 3D arrangement of A (**B**). This picture shows the generation of clusters in the presence of nickel. The nickel clusters were much smaller compared to the nickel oxide and had an average diameter of 50 ± 10 μm (**C**). The 3D arrangement of C (**D**). statistical analysis of total cell layer thickness. This graph shows the thickness of cell layer (live and dead) of *P. aeruginosa* that are surrounded nickel oxide NPs cluster (**E**). Statistical analysis of the formed biofilm (eDNA). This graph shows the thickness of formed biofilm of *P. aeruginosa*, which are surrounding nickel oxide cluster (**F**). Statistical analysis of diagrams (**E**,**F**) shows that the thickness of the cell layer is mostly in the range of 30 to 40 µm. The thickness of the biofilm is mostly in the range of 20 to 30 µm. Here, the *X*-axis represents the thickness of the cell layer or biofilm and the *Y*-axis shows the relevant percentage. Definition of statistical significance: GP: 0.1234 (ns), 0.0332 (*), 0.0021 (**), 0.0002 (***). Scalebar = 200 μm.

**Table 1 microorganisms-10-02220-t001:** The list of tested metal particles. This table presents the tested metal particles with the indicated sizes given by the manufacturers; the minimum inhibitory concentration assay was performed on *P. aeruginosa* in our laboratory and the results are presented as MIC in the table. All nanoparticles were purchased from Sigma, with the exception of silver microparticles, which were from Biogate (Bremen, Germany).

Nano-Metal	Size	MIC
Zinc	40–60 nm	25 mM
Zinc oxide	<100 nm	50 mM
Copper	<500 nm	25 mM
Titanium oxide	<25 nm	-
Molybdenum	<100 nm	-
Nickel	<100 nm	10 mM
Nickel oxide	<50 nm	50 mM
Tungsten	10 µm	-
Tungsten oxide	<100 nm	-
Silver (nano)	<100 nm	25 mM
Silver (micro)	10 µm	25 mM

**Table 2 microorganisms-10-02220-t002:** Comparison of particle sizes between the particle sizes given by the manufacturers and our measurements. Our observations show that the input size of nickel given by the manufacturer does not correlate with our measurements.

Title 1	Formula	Size (Company)	Size TEM
Nickel	Ni	<100 nm	<300 nm
Nickel oxide	NiO	<50 nm	<50 nm

## Data Availability

All presented data generated by our experoments are included in the manuscript.

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
