# Peer review of "Pseudomonas aeruginosa Clusters Toxic Nickel Nanoparticles to Enhance Survival"

_microorganisms, 2022, doi:10.3390/microorganisms10112220_

Round 1

Reviewer 1 Report

In this manuscript, the effects of transition metal nanoparticles on the growth and biofilm formation of two bacteria (S. aureus and P. aeruginosa) are investigated. Different concentrations of nickel and nickel oxide were used to determine toxicity. The results are interesting and unique, P. aeruginosa presents a particular defense mechanism against toxic nanoparticles by clustering the nanoparticles in biofilm.  I agree with publishing after minor revisions.

The introduction is well documented, the authors of the manuscript citing the latest research on the antibacterial potential of these Ni and NiO nanoparticles.

Line number

44. neurohabilitation should be Neurohabilitation

46. the various .....The various

56. in the.......In the

79. in addition.....In addition

85. also.....Also  

Is water decontaminated in this way not toxic to consumers?

Reviewer 2 Report

The manuscript should be conducted some revisions befoer accepting to publish.

1. The more related and key result data should be added in the abstract section. The name of microorgnaism should be used the Italic type. 

2. The number of paragraphes should be reduced and some related paragrahpes can be combined.

3. The data analysis method should be added at the end of materials and methods section.

4. The more related articles can be cited and discussed in the discussed setion to explane the effect mechanism.

5. The English language should be revised in the whole manucript.

Reviewer 3 Report

Comments and Suggestions

The present manuscript investigated the capability of P. aeruginosa to  filter nickel oxide from the medium by biofilm formation allowing the bacteria to continue living without contact to the

stressor. In my opinion this is an interesting topic due to the increasing research on nanoparticles, but data suffer from poor description overall and are not adequately analyzed.

 In the introduction it is not sufficiently clear why this is an interesting subject of study and in the discussion what this paper adds to the existing literature it is not clear too. Because of important incorrectness in methodology and results I do not recommend it for publications.

Here is some comments and suggestions:

1)      There are some difficulties in several parts of the manuscript to understand what is meant by the authors, and thus the manuscript needs to be edited by a person whose mother language is English.

2)      There are some typos (italics and capital letters..)

3)      The purpose of the paaper is confusing and repetitive Lines 87-105

4)      How you identify a microorganism isolated from domestic washing machines as P. aeruginosa? Is it Gram-positive?

5)      Please explain more clearly how the collected isolates are maintained in laboratory and propagated for the experiments .

6)      Lines 109-11 please is missing the grown in appropriate culture media for the two bacteria and the biofilm formation ability for P. aeruginosa. Why the 11 reference? It is not appropriate

7)      Lines 114-117 : authors say to buy two nanoparticles, in the table 1 are described 11; which are manufacturing data?

8)      Lines 126-131 why “maximum Inhibitory Concentration”, is it IC50? Please show the referce method used; Why in the results is reported MIC; please explain medium used for CFU

9)      Lines 133-141 “time kill assay” is confusing

10)  Lines 195-196 and table  “ several NPs to investigate the toxicity of these metals”: it is the first time that you talk about other nanoparticles beyond Nickel  and Nickel oxide. Table 1 is rambling.

11)  Are tests made in duplicate or in triplicate?

12)  Results and discussion require an important improvement especially in statistical analysis An important improvement is requested especially in statistical analysis for all figures

13)  Despite the large amount of work, the authors should better discuss the collected results because in some points they generate confusion.

Round 2

Reviewer 3 Report

Accept after minor revision (corrections to minor methodological errors and text editing